# Partial Substitution of Whey Protein Concentrate with Spray–Dried Porcine Plasma or Soy Protein Isolate in Milk Replacer Differentially Modulates Ileal Morphology, Nutrient Digestion, Immunity and Intestinal Microbiota of Neonatal Piglets

**DOI:** 10.3390/ani13213308

**Published:** 2023-10-24

**Authors:** Yuwei Zhang, Qiang Zhou, Shiya Liu, Xiang Quan, Zhengfeng Fang, Yan Lin, Shengyu Xu, Bin Feng, Yong Zhuo, De Wu, Lianqiang Che

**Affiliations:** Key Laboratory for Animal Disease-Resistant Nutrition of China Ministry of Education, Institute of Animal Nutrition, Sichuan Agricultural University, Chengdu 611130, China; zyuwei0206@163.com (Y.Z.); 18227553406@163.com (Q.Z.); lsy070766@163.com (S.L.); xiangquan1027@163.com (X.Q.); zfang@sicau.edu.cn (Z.F.); linyan936@163.com (Y.L.); shengyu_x@hotmail.com (S.X.); fengbin@sicau.edu.cn (B.F.); zhuoyong@sicau.edu.cn (Y.Z.); wude@sicau.edu.cn (D.W.)

**Keywords:** piglets, milk replacer, dietary protein, intestinal function, digestive enzymes, microbiome

## Abstract

**Simple Summary:**

Milk replacer is recommended as an effective substitute when sow milk is insufficient or unavailable to meet the requirements of neonates, and the protein sources of milk replacer are vital for the intestinal maturation and health of neonates. In this study, we investigate the effects of SDPP or SPI partially substituting WPC in milk replacer on the growth performance, ileal morphology, nutrient digestion, immunity and intestinal microbiota of neonatal piglets. Our results confirm the efficacy of SDPP in milk replacer for improving the growth and intestinal health of neonatal piglets.

**Abstract:**

Appropriate protein sources are vital for the growth, development and health of neonates. Twenty–four 2–day–old piglets were randomly divided into three groups and fed isoenergetic and isonitrogenous diets. The experimental diets included a milk replacer with 17.70% whey protein concentrate (WPC group), a milk replacer with 6% spray–dried porcine plasma isonitrogenously substituting WPC (SDPP group), and a milk replacer with 5.13% soy protein isolate isonitrogenously substituting WPC (SPI group). Neonatal piglets were fed milk replacer from postnatal day 2 (PND 2) to day 20 (PND 20). The growth performance, intestinal morphology, activities of digestive enzymes, plasma biochemical parameters, immunity–related genes, short–chain fatty acids (SCFA) and intestinal microbiota in the colonic chyme were determined. The results showed that SDPP–fed piglets had higher final BW (*p* = 0.05), ADG (*p* = 0.05) and F/G (*p* = 0.07) compared with WPC– and SPI–fed piglets, and SDPP–fed piglets had a lower diarrhea index (*p* < 0.01) from PND 2 to PND 8. SDPP–fed piglets had an increased ileal villus height (*p* = 0.04) and ratio of villus height to crypt depth (VCR) (*p* = 0.02), and increased activities of sucrase (*p* < 0.01), lactase (*p* = 0.02) and trypsin (*p* = 0.08) in the jejunum, compared with WPC– and SPI–fed piglets. Furthermore, SPI–fed piglets had an increased mRNA expression of *IL-6* (*p* < 0.01) and concentration of plasma urea (*p* = 0.08). The results from LEfSe analysis showed that SDPP–fed piglets had a higher abundance of beneficial *Butyricicoccus* compared with WPC– and SPI–fed piglets, in which higher abundances of pathogenic bacteria such as *Marinifilaceae*, *Fusobacterium* and *Enterococcus* were observed. Moreover, SDPP–fed piglets had an increased concentration of butyric acid (*p* = 0.08) in the colonic chyme compared with WPC– and SPI–fed piglets. These results suggest that neonatal piglets fed milk replacer with SDPP partially substituting WPC had improved growth performance and intestinal morphology and function, associated with higher digestive enzyme activity and fewer pathogenic bacteria.

## 1. Introduction

Genetic selection for highly prolific sows has been found to largely increase litter size [1]. However, it is challenging for sows to nurse a number of piglets that exceeds the number of functional teats, which results in higher pre-weaning mortality and poor pig uniformity at weaning [2]. Milk replacer is recommended as an effective dietary strategy when sow milk is unavailable or insufficient to satisfy piglets’ nutritional needs [3]. Studies on optimizing milk replacer using functional additives, such as oligosaccharide [4,5], lactoferrin [6] and β-glucan [7], have been reported, but limited attention has been given to protein sources. Protein is a crucial nutrient for neonates, serving as a vital source of essential amino acids and fundamental bioactive substances for early-stage development [8]. Meanwhile, the digestion and absorption of protein are closely associated with nutritional status and intestinal health [9].

As a dairy–based protein, whey protein concentrate (WPC) has been widely used as a major protein source in milk replacer [10,11]. It has been reported that milk replacer with WPC contributes to the intestinal maturation and health of preterm piglets [12]. As an expensive component, however, dairy-based protein has been partially substituted with wheat or soy proteins in milk replacers for calves [13]. And soy protein–based formula has been reported to maintain the normal growth and development of infants and reduce gastrointestinal symptoms in infants who are intolerant to cows’ milk [14,15,16]. Although soy protein isolate (SPI) is considered an ideal plant–based protein in formula milk, there are concerns regarding the potential negative effects of soy–based protein on digestibility and immune and nervous system development due to insufficient processing for anti-nutritive factors [17,18,19]. To our knowledge, however, very limited data are available on the effects of soy–based protein as a substitute for dairy–based protein in milk replacer on the growth and development of piglets. In contrast, functional proteins could also be considered for improving neonatal growth and maturity. Spray–dried porcine plasma (SDPP) is a high–quality animal–based protein, and is widely used to improve the intestinal health, immunity and growth of weaning piglets due to its enrichment in fibrinogen, immunoglobulins and albumin [20,21,22]. However, the feeding effect of SDPP as a protein source to partially substitute WPC in milk replacer for piglets has not been determined.

In this study, therefore, we aimed to investigate the effects of partially substituting the dairy–based protein WPC in milk replacer with the animal–based protein SDPP or the plant–based protein SPI on the growth performance, intestinal morphology, activity of digestive enzymes, immunity–related parameters, intestinal microbiota composition and metabolites in neonatal piglets.

## 2. Materials and Methods

### 2.1. Experimental Design, Animals and Growth Performance

Twenty–four Landrace/Yorkshire crossbred piglets (2 days old) with an average initial body weight of 1.55 ± 0.23 kg, including a mix of females and males, were selected for this experiment. After birth, piglets were allowed to suckle from their respective sows for 48 h prior to being transported to a nursery room, and were housed individually in stainless steel metabolic crates (0.6 × 0.9 × 0.6 m). Piglets were randomly divided into three treatment groups receiving experimental diets, including a milk replacer with 17.70% WPC (WPC group, *n* = 8), a milk replacer with 6% SDPP isonitrogenously substituting WPC (SDPP group, *n* = 8), and a milk replacer with 5.13% SPI isonitrogenously substituting WPC (SPI group, *n* = 8). The nursery room temperature was controlled at 28~31 °C. The intake of milk replacer was converted into dry matter intake using a dry matter–to–water ratio of 1:4, and piglets were weighed individually at the start and end of the study to calculate the average daily feed intake (ADFI), average daily gain (ADG) and ratio of ADFI to ADG (F/G). Visual diarrhea scores were assessed three times daily and the diarrhea index was calculated [23].

### 2.2. Milk Replacer

The milk replacer’s ingredients and nutrient levels are described in Table 1, and it was formulated to mimic the macronutrient composition of sow milk [24,25,26]. The milk replacer was mixed with 40 °C water at a ratio of 1:4 and added to a milk bucket for storage. It was added 7 times a day to ensure that there was surplus milk in the bucket. The milk was transported from the bucket to the nipple on the metabolism cage, and the piglets could freely drink milk through the nipple until sacrifice on PND 21.

### 2.3. Sample Collection

Prior to slaughter, milk replacer and water were withheld from piglets for 12 h. Once their final body weight was recorded, blood was collected from the anterior vena cava. Plasma samples were obtained by centrifuging blood samples stored in heparin anticoagulated tubes at 3000× *g* for 15 min at 4 °C. In order to conduct histological analysis, piglets were anesthetized intramuscularly using 0.5 mL of xylazine and midazolam. Following slaughter, the intestines of piglets were isolated and a 2 cm length piece of the ileum was stored in a 4% paraformaldehyde solution. Tissue samples from the jejunum and ileum were collected and washed in cold saline solution (NaCl 9 g/L, 4 °C). Similarly, chyme samples from the colon were collected, frozen in liquid nitrogen and stored at −80 °C for 16S rRNA sequencing to assess the microbial community composition.

### 2.4. Histomorphology

Pieces of ileum were fixed in 4% paraformaldehyde solution for histomorphometric analysis. Ileum samples were dehydrated and embedded in paraffin. Sections 5 μm thick, cut using a Leica RM2235 microtome (Leica Microsystems Ltd., Shanghai, China), were stained using the periodic acid–Schiff (PAS) method. Image Pro Plus 6.0 software (Media Cybernetics, Rockville, MD, USA) was used to acquire images and to measure the villus height, the depth of crypts and the area of goblet cells in the tissue.

### 2.5. Digestive Enzyme Activity

Jejunum samples were homogenized using 4 °C pre–cooled 0.9% saline for further analysis. The protein content of the samples was determined using the Coomassie brilliant blue method, and the test kit was purchased from Nanjing Jiancheng (Nanjing Jiancheng Bioengineering Institute, Nanjing, China). The activities of sucrase, maltase, lactase and trypsin were determined according to the instructions for the Nanjing Jiancheng kit (Nanjing Jiancheng Bioengineering Institute, Nanjing, China). The optical density values of disaccharidase and trypsin were measured at 505 nm and 253 nm, respectively, using a SpectraMax190 microplate reader (Molecular Devices Corporation, San Jose, CA, USA); then, the activities of the enzymes were calculated according to the change in absorbance.

### 2.6. Gene Expressions

Extracted ileal RNA with a concentration of 500~1000 ng/μL was used to synthesize cDNA, according to the instructions for the reverse transcription kit (Vazyme) (RNase-free ddH_2_O, 4× gDNA Wiper Mix, 5× HiScript III qRT SuperMix). Quantitative PCR (2× ChamQ Universal SYBR qPCR Master Mix, Primer, Template DNA/cDNA, ddH_2_O) was carried out using the QuantStudio5 instrument (Thermo Fisher Scientific, Waltham, MA, USA). Data were normalized using β-actin and the relative expression was calculated using the 2^−ΔΔCt^ method [27]. The primers used in this experiment are listed in Table 2.

### 2.7. 16S rRNA Amplicon Sequencing

Colonic chyme samples were used for 16S rRNA amplicon (Novogene Technology Co., Ltd., Beijing, China). Genomic DNA was extracted using either the CTAB or SDS method. Specific primers with barcodes, the Phusion^®^ High–Fidelity PCR Master Mix with GC buffer (New England Biolabs) and high–efficiency high–fidelity enzymes were used to perform PCR. The NEBNext^®^ Ultra™ IIDNA Library Prep Kit (New England Biolabs, Ipswich, MA, USA) was utilized for library construction. Then, the constructed library was qualified and sequenced using the NovaSeq6000 instrument (Illumina, San Diego, CA, USA). After careful processing and analysis, the effective tags were obtained and denoised using QIIME2 software (2019.1) to generate ASVs (Amplicon Sequence Variants) and feature tables. The species information for each ASV was determined by comparing it to a database. Finally, QIIME2 software was used to conduct alpha and beta diversity analysis.

### 2.8. SCFA Analysis

Briefly, 0.5 g of colonic chyme sample and 1.2 mL of ultrapure water were added to a centrifuge tube after thawing, allowed to stand for 30 min and centrifuged at 10,000× *g* for 15 min. After centrifugation, 0.2 mL of 25% (*w*/*v*) metaphosphoric acid and 23.3 μL of 210 mmol/L crotonic acid were added to 1 mL of supernatant, mixed well, and left to stand at 4 °C for 30 min, and then, centrifuged at 8000× *g* for 10 min. After centrifugation, 0.9 mL of chromatographic methanol (1:3 dilution) was added to 0.3 mL of supernatant, and mixed well. Finally, the supernatant was filtered using a 0.22 μm filter membrane into a 1.5 mL EP tube for later use. SCFA concentrations were measured using GC CP3800 gas chromatography (Varian Medical Systems, Palo Alto, CA, USA).

### 2.9. Statistical Analysis

The Shapiro–Wilk test and UNIVARIATE procedures of SAS 9.4 (SAS Institute, Inc., Cary, NC, USA) were used to analyze variance homogeneity and normality, respectively. Data were analyzed using one–way analysis of variance (ANOVA), and the LSD method was used for multiple comparisons. Data are presented as mean ± pooled standard error (SEM) and considered significant at *p* < 0.05 and a tendency at *p* < 0.10.

## 3. Results

### 3.1. Growth Performance

SDPP–fed piglets tended to have a higher final BW (*p* = 0.05) and ADG (*p* = 0.05), and lower F/G (*p* = 0.07), compared with WPC– and SPI–fed piglets (Table 3). SDPP–fed piglets had significantly reduced diarrhea indexes from postnatal day 2 (PND 2) to day 8 (PND 8) compared with WPC– and SPI–fed piglets (Figure 1). No significant difference in ADFI was observed across the treatment groups.

### 3.2. Ileal Morphology

SDPP–fed piglets had a significantly increased villus height in the ileum compared with WPC– and SPI–fed piglets (Figure 2b). SDPP–fed piglets had a significantly increased ratio of villus height to crypt depth (VCR) compared with WPC–fed piglets (Figure 2b). No significant differences in ileal crypt depth and goblet cell density were observed across the treatment groups.

### 3.3. Activities of Enzymes

Both WPC– and SDPP–fed piglets had significantly increased activities of sucrase and lactase in the jejunum compared with SPI–fed piglets (Figure 3a,c). SDPP–fed piglets tended to have higher trypsin activity compared with WPC– and SPI–fed piglets (*p* = 0.08, Figure 3d). No significant difference in the activity of maltase was observed across the treatments.

### 3.4. Relative mRNA Expressions of Barrier-Function- and Inflammation-Related Genes

SPI–fed piglets had significantly increased mRNA expressions of *IL–6* and *IL–10* in the ileum compared with WPC– and SDPP–fed piglets (Figure 4b). No significant differences in the mRNA expressions of *Occludin*, *ZO–1*, *Ocaudin–1*, *TNF–α* and *IL–1β* were observed across the treatments (Figure 4a,b).

### 3.5. Plasma Biochemical Parameters

SPI–fed piglets tended to have a higher urea concentration compared with WPC– and SDPP–fed piglets (*p* = 0.08, Table 4). No significant differences in the concentrations of C3, IgG, GLU, TC, LDL–C, HDL–C, NEFA and TG were observed across the treatments.

### 3.6. Colonic Microbiome

Venn analysis identified 1244, 1104 and 1219 unique OTUs in the WPC, SDPP and SPI groups, respectively (Figure 5a, Table 5). *Firmicutes*, *Bacteroidetes* and *Proteobacteria* were the three most abundant phyla (Figure 6a). SDPP–fed piglets had lower *Proteobacteria* (7.73%) compared with WPC– (21.08%) and SPI–fed piglets (28.55%).

LEfSe (LDA Effect Size) is used to find biomarkers with statistical differences across groups. The most abundant phylotypes contributing to the differences across the intestinal microbiota of the WPC, SDPP and SPI groups were found to be *f__Marinifilaceae*, *f__Rikenellaceae*, *g__JG30_KF_CM66*, *f__Anaerovoracaceae* and *g__A21b* in the WPC group, *g__Butyricicoccus* in the SDPP group, and *g__Fusobacterium*, *g__Enterococcus*, *g__Sumerlaea* and *f__Rubritaleaceae* in the SPI group (Figure 7).

### 3.7. SCFA Concentrations

SDPP–fed piglets tended to have a higher concentration of butyric acid in the colonic chyme compared with WPC– and SPI–fed piglets (*p* = 0.08, Table 6). No significant differences in concentrations of acetic acid, propionic acid, isobutyric acid, isovaleric acid, valeric acid and total SCFA were observed across the treatments.

## 4. Discussion

Whey protein concentrate (WPC) is a commonly used dairy–based protein in milk replacer for piglets [28]. However, the efficacy of the partial substitution of WPC with animal–based or plant–based protein on the growth and intestinal function of piglets needs to be determined.

In the present study, the higher final BW and ADG, and the lower F/G, in SDPP–fed piglets indicated the crucial role of SDPP in improving the growth performance of piglets, which is consistent with previous results from studies that include SDPP in weaning diets in which it increased post–weaning growth rate [29,30,31]. Also, calves fed a milk replacer using SDPP substituting 20% of the WPC had significantly increased feed intake and feed conversion ratios [32]. Meanwhile, the diarrhea index from PND 2 to PND 8 was lower in SDPP–fed piglets, which is consistent with a previous study in which the fecal scores of calves tended to be lower when SDPP was included in the milk replacer [32]. The improvements in SDPP on growth rate and diarrhea index could be associated with improved intestinal function. The intestinal mucosa plays a crucial role in nutrient absorption [33]. Our results showed that SDPP–fed piglets had significantly increased ileal villus height and VCR compared with WPC– and SPI–fed piglets, indicating an expansion of the intestinal epithelial surface area [34], and an improvement in epithelial cell renewal [35,36]. Consistently, the weaned piglets fed an SDPP diet had significantly increased ileal villus height relative to those fed a soybean protein concentrate (SPC) diet [37]. Intestinal enzyme activity can be altered in response to diet as early as the second day after birth [38]. Piglets fed colostrum-containing plasma showed increased activities of sucrase and maltase (50% and 200%, respectively) compared with colostrum-fed piglets [39]. In this study, moreover, SDPP–fed piglets had increased activities of sucrase, lactase and trypsin in the jejunum compared with WPC– and SPI–fed piglets, which suggested that SDPP–fed piglets were favorable for protein and carbohydrate digestion.

In contrast, SPI–fed piglets had the lowest growth rate, which may be associated with poor digestive capability, immunity and bacterial structure. Our study found that SPI–fed piglets had higher plasma urea, indicating a potential increase in protein breakdown or lower protein utilization, because urea is the main end product of amino acid and protein metabolism, which is negatively correlated with amino acid utilization and protein retention [40]. Immunity is crucial for the intestinal health of neonates. The dynamic changes in levels of pro–inflammatory factor *IL–6* and anti-inflammatory factor *IL–10* play an important role in regulating the inflammatory response system. In this study, both *IL–6* and *IL–10* were over-expressed in the ileum of SPI–fed piglets. This finding seems paradoxical considering the anti–inflammatory effects of *IL–10*. However, previous research has found that an increased level of *IL–10* is a protective mechanism for the body against the release of a large number of pro–inflammatory cytokines [41]. In addition, elevated *IL–6* can also stimulate the release of *IL–10* in local tissues [42]. Therefore, we speculated that feeding milk replacer including SPI may pose a challenge to the intestinal immunity of neonatal piglets.

The normal intestinal microbiota is essential for intestinal homeostasis [43]. Dietary protein is the preferred substrate for the intestinal microbiota and affects the composition and metabolic activity of microbiota [44]. In this study, LEfSe analysis found that microbiota composition was significantly different across the WPC, SDPP and SPI groups. At the phylum level, *Firmicutes*, *Bacteroidetes* and *Proteobacteria* were the most abundant bacterial phyla in the colonic chyme. SDPP-fed piglets had lower *Proteobacteria* levels compared with WPC– or SPI–fed piglets. Interestingly, previous research demonstrated that breast–fed infants had lower *Proteobacteria* levels (3.29%) compared with formula–fed infants (13.85%) [45]. Furthermore, *Proteobacteria* has been linked to inflammation and intestinal dysbiosis [46,47]. In this study, moreover, the abundance of *Butyricicoccus*, a high butyrate–producing probiotic [48], was increased in SDPP–fed piglets, which also explains the increased concentration of butyrate in SDPP–fed piglets. As a short–chain fatty acid, butyrate not only provides energy for colonic cells, but also inhibits the expression of pro–inflammatory cytokines [49]. In addition, the abundance of some pathogenic bacteria, such as *Marinifilaceae*, *Fusobacterium* and *Enterococcus*, was increased in the colonic chyme of WPC– and SPI–fed piglets. It has been reported that *Marinifilaceae* was enriched in the lumen of colitis and infection animal models [50,51]. One member of *Fusobacterium*, *F. nucleatum* has been recognized as an opportunistic pathogen that plays an important role in intestinal diseases, which promotes the inflammatory response and induces the release of the pro–inflammatory cytokines *IL–6* and *IL–8* [52,53,54]. This finding may explain the over-expression of *IL–6* in SPI–fed piglets.

## 5. Conclusions

The use of SDPP to partially substitute WPC in milk replacer improved the growth performance and intestinal function of piglets, associating with increased activities of digestive enzymes, decreased expressions of inflammatory genes, and the lower presence of pathogenic bacteria compared with the WPC and SPI groups. However, further research is needed to investigate the long-term implications of these outcomes.

## Figures and Tables

**Figure 1 animals-13-03308-f001:**
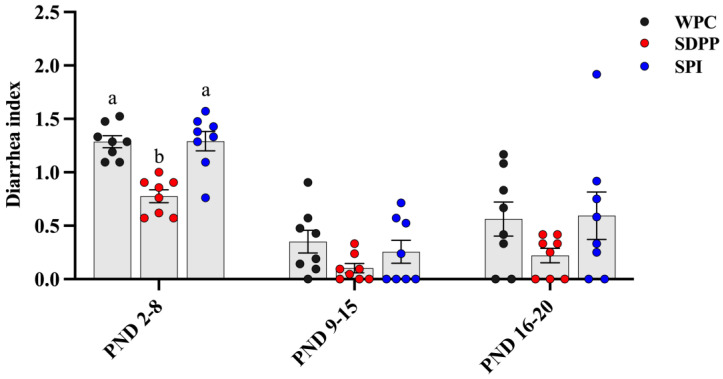
Effect of protein sources of milk replacer on diarrhea index of piglets. PND, postnatal day. Values are presented as mean ± standard error (*n* = 8). ^ab^ Means marked with the same letter do not significantly differ (*p* > 0.05).

**Figure 2 animals-13-03308-f002:**
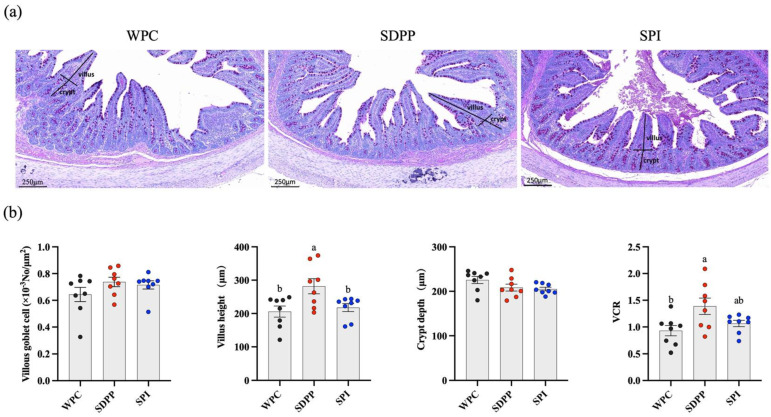
Effect of protein sources of milk replacer on ileal morphology of piglets. Values are presented as mean ± standard error (*n* = 8). (**a**) Representative images of ileum villus height and crypt depth in WPC–, SDPP– and SPI–fed piglets. The sections were stained using the periodic acid–Schiff (PAS) staining method. (**b**) Ileum statistical analysis for goblet cell density, villus height, crypt depth and VCR. ^ab^ Means marked with the same letter do not significantly differ (*p* > 0.05).

**Figure 3 animals-13-03308-f003:**
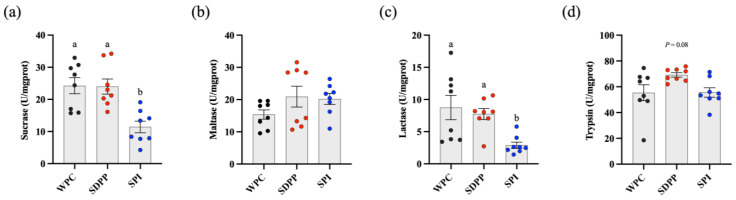
Effect of protein sources of milk replacer on activities of trypsin and disaccharidase across the jejunum. (**a**) Sucrase; (**b**) maltase; (**c**) lactase; (**d**) trypsin. Values are presented as mean ± standard error. ^ab^ Means marked with the same letter do not significantly differ (*p* > 0.05).

**Figure 4 animals-13-03308-f004:**
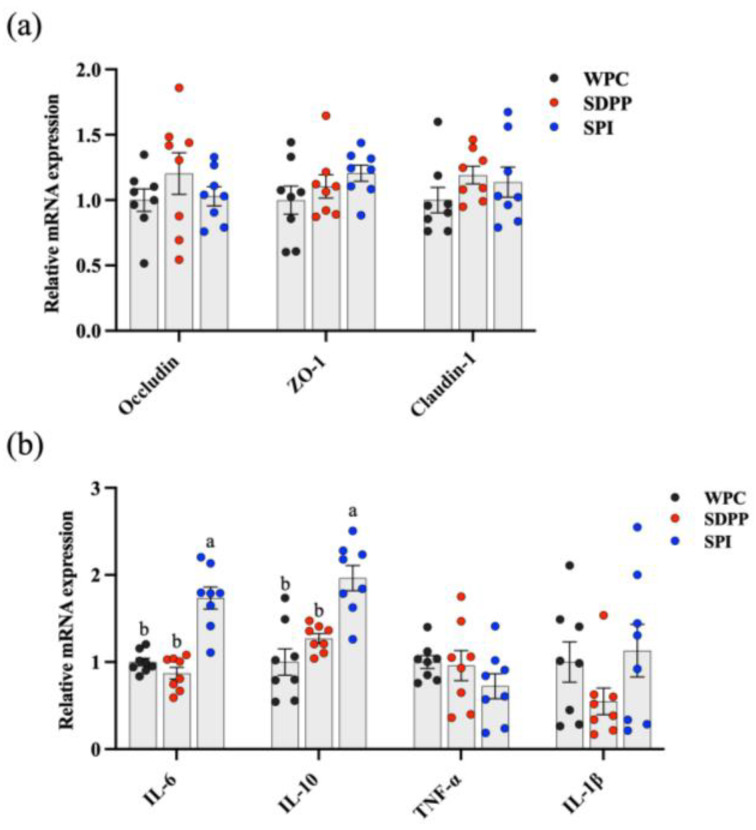
The relative mRNA expressions of barrier–function– and inflammation–related genes in the ileum. (**a**) Relative mRNA expression of barrier genes; (**b**) relative mRNA expression of inflammatory genes. Values are presented as mean ± standard error (*n* = 8). ^ab^ Means marked with the same letter do not significantly differ (*p* > 0.05).

**Figure 5 animals-13-03308-f005:**
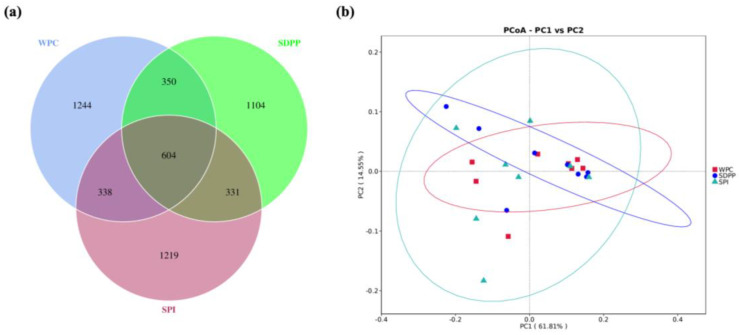
Comparison of colon microbiomes. (**a**) Venn diagram; (**b**) weighted UniFrac principal co–ordinate analysis (PCoA).

**Figure 6 animals-13-03308-f006:**
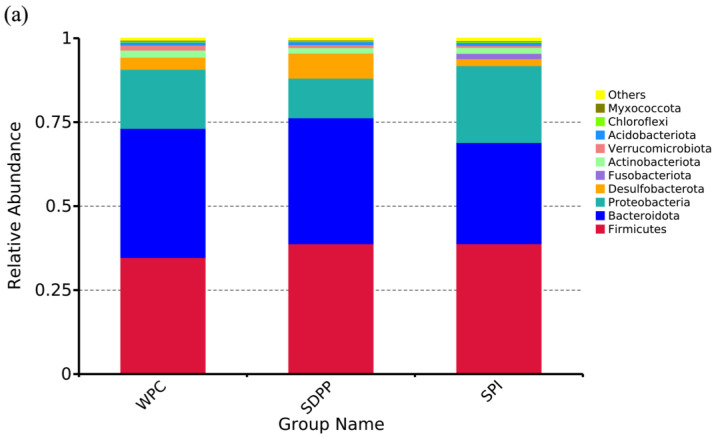
Relative abundance of colon microbiome at phylum (**a**) and genus (**b**) level in WPC–, SDPP– and SPI–fed piglets.

**Figure 7 animals-13-03308-f007:**
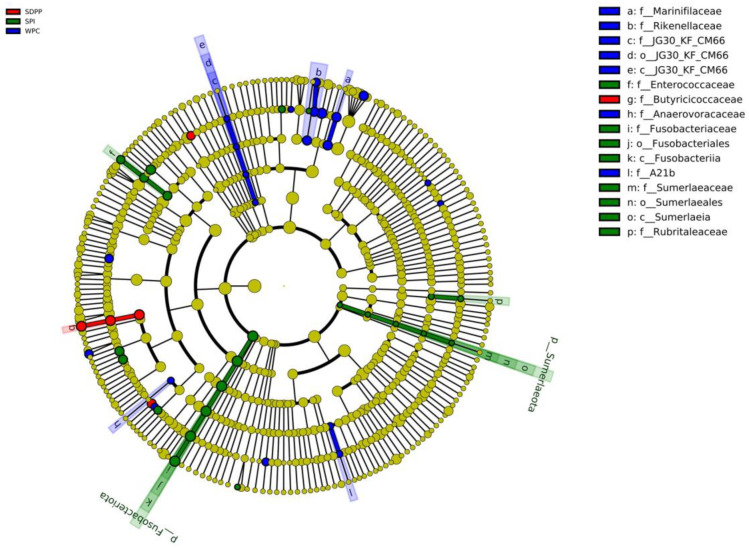
Biomarkers with statistically significant differences across groups were found via LEfSe analysis, illustrated in an evolutionary branch diagram.

**Table 1 animals-13-03308-t001:** Ingredients and nutrient levels for neonatal piglets fed milk replacers from PND 2 to PND 20.

	WPC	SDPP	SPI
**Ingredients**	%	%	%
Whey powder	12.17	11.18	12.95
Whole milk powder	10.00	10.00	10.00
Skim milk powder	24.00	24.00	24.00
Whey protein concentrate, 73% CP ^1^	17.70	11.92	11.05
Plasma protein powder, 72% CP ^2^	-	6.00	-
Soy protein isolate, 83% CP ^3^	-	-	5.13
Oil powder	25.00	25.70	25.00
Sucrose	5.00	5.00	5.00
L-lysine hydrochloride	-	0.10	0.36
DL-methionine	0.34	0.37	0.50
L-threonine	-	-	0.15
L-tryptophan	-	0.02	0.05
Calcium hydrogen phosphate	1.94	1.48	1.84
Calcium formate	0.20	0.58	0.32
Choline chloride, AR	0.08	0.08	0.08
Vitamin premix ^4^	0.03	0.03	0.03
Mineral premix ^5^	0.02	0.02	0.02
Antioxidants	0.02	0.02	0.02
Citric acid	3.50	3.50	3.50
Total	100.00	100.00	100.00
**Nutrient levels** **, calculated**			
Digestible energy, Mcal/kg	4.26	4.26	4.26
Crude protein, %	25.02	25.03	25.03
Ether extract, %	16.13	16.42	16.07
Lactose, %	28.02	27.03	28.19
Calcium, %	1.00	1.00	1.00
Total phosphorus, %	0.71	0.71	0.71
Total lysine, %	2.56	2.56	2.56
Total methionine, %	0.95	0.91	1.03
Total methionine + cystine, %	1.42	1.42	1.42
Total threonine, %	1.57	1.58	1.57
Total tryptophan, %	0.46	0.46	0.46

^1^ WPC, DE: 4.17 Mcal/kg, CP: 73% (Fonterra Co–operative Group, Ltd., Auckland, New Zealand). ^2^ SDPP, DE: 3.74 Mcal/kg, CP: 72% (American Protein Corporation., Ankeny, IA, USA). ^3^ SPI, DE: 4.15 Mcal/kg, CP: 83% (Mountain Pine Biological Products Co., Ltd., Linyi City, China). ^4^ Provided per kg of diet: VA 15,000 IU; VD_3_ 5000 IU; VE 40 IU; VK_3_ 5.00 mg; VB_1_ 5.00 mg; VB_2_ 12.50 mg; VB_6_ 6.00 mg; VB_12_ 0.06 mg; D–biotin 0.25 mg; D–pantothenic acid 25.00 mg; folic acid 2.50 mg; nicotinamide 50.00 mg. ^5^ Provided per kg of diet: Zn 100 mg; Cu 6 mg; Fe 100 mg; Mn 4 mg; I 0.14 mg; Se 0.3 mg.

**Table 2 animals-13-03308-t002:** Primer sequences of the target and reference genes.

Genes	Primer Sequence (5′–3′)	Product (bp)	GenBank Accession
*IL–6*	F: TGCAGTCACAGAACGAGTGG	116	NM_214399.1
	R: CAGGTGCCCCAGCTACATTAT		
*IL–10*	F: AATCTGCTCCAAGGTTCCCG	224	NM_214041.1
	R: TGAACACCATAGGGCACACC		
*IL–1β*	F: TCTGCCCTGTACCCCAACTG	64	NM_214055.1
	R: CCAGGAAGACGGGCTTTTG		
*TNF–a*	F: CCACGTTGTAGCCAATGTCA	395	NM_214022.1
	R: CAGCAAAGTCCAGATAGTCG		
*Occludin*	F: CAGCAGCAGTGGTAACTTGG	110	NM_001163647.2
	R: CCGTCGTGTAGTCTGTCTCG		
*ZO–1*	F: CCGCCTCCTGAGTTTGATAG	189	XM_013993251.1
	R: CAGCTTTAGGCACTGTGCTG		
*Claudin–1*	F: ACTGGCTGGGCTGCTGCTTCTCT	101	NM_001244539.1
	R: GGATAGGGCCTTGGTGTTGGGTAA		
*β–actin*	F: GGCGCCCAGCACGAT	66	XM_021086047.1
	R: CCGATCCACACGGAGTACTTG		

*IL-6*, interleukin 6; *IL-10*, interleukin 10; *IL-1β*, interleukin-1β; *TNF-α*, tumor necrosis factor-α; *ZO-1*, tight junction protein 1.

**Table 3 animals-13-03308-t003:** Effect of protein sources of milk replacer on growth performance of piglets (*n* = 8).

		Diets		SEM	*p*-Value
	WPC	SDPP	SPI		
Initial BW (kg)	1.54	1.56	1.56	0.08	0.98
Final BW (kg)	3.39	4.37	3.04	0.37	0.05
ADG (g/d)	103	157	83	20.86	0.05
ADFI (g/d)	130	168	140	13.04	0.12
F/G	1.72	1.16	2.27	0.36	0.07

WPC, milk replacer with 17.70% whey protein concentrate (WPC); SDPP, milk replacer with 6% spray-dried porcine plasma (SDPP) isonitrogenously substituting WPC; SPI, milk replacer with 5.13% soy protein isolate (SPI) isonitrogenously substituting WPC. ADG, average daily gain; ADFI, average daily feed intake; F/G, ratio of ADFI to ADG.

**Table 4 animals-13-03308-t004:** Effect of protein sources of milk replacer on plasma biochemical parameters of piglets (*n* = 8).

		Diets		SEM	*p*-Value
	WPC	SDPP	SPI		
C3, mg/L	0.09	0.09	0.09	0.01	0.81
IgG, g/L	1.86	1.83	1.88	0.06	0.81
Urea, mmol/L	15.71	13.21	24.64	3.61	0.08
GLU, mmol/L	2.80	2.44	2.89	0.83	0.93
TC, mmol/L	3.18	3.40	3.62	0.29	0.56
LDL–C, mmol/L	1.36	1.49	1.71	0.18	0.37
HDL–C, mmol/L	0.81	0.74	0.67	0.06	0.25
NEFA, µmol/L	386.12	320.08	341.02	76.63	0.82
TG, mmol/L	0.76	0.88	0.91	0.16	0.72

WPC, milk replacer with 17.70% whey protein concentrate (WPC); SDPP, milk replacer with 6% spray–dried porcine plasma (SDPP) isonitrogenously substituting WPC; SPI, milk replacer with 5.13% soy protein isolate (SPI) isonitrogenously substituting WPC. C3, complement 3; IgG, immunoglobulin G; GLU, glucose; TC, total cholesterol; LDL–C, low–density lipoprotein cholesterol; HDL–C, high–density lipoprotein cholesterol; TG, triglyceride.

**Table 5 animals-13-03308-t005:** Effect of protein sources of milk replacer on α–diversity index of piglets (*n* = 8).

		Diets		SEM	*p*-Value
	WPC	SDPP	SPI		
Shannon	5.29	4.98	5.17	0.18	0.52
Simpson	0.90	0.89	0.89	0.02	0.46
Dominance	0.10	0.11	0.11	0.02	0.46
Pielou_e	0.59	0.56	0.57	0.02	0.62
Chao1	534.32	492.19	541.98	29.13	0.44

**Table 6 animals-13-03308-t006:** Effect of protein sources of milk replacer on SCFA concentrations in colonic chyme of piglets (*n* = 8).

		Diets		SEM	*p*-Value
	WPC	SDPP	SPI		
Acetic acid, μmoL/g	31.48	29.75	31.33	2.12	0.81
Propionic acid, μmoL/g	11.61	12.17	11.29	0.50	0.47
Isobutyric acid, μmoL/g	1.06	0.99	1.07	0.07	0.69
Butyric acid, μmoL/g	3.33	4.32	2.86	0.45	0.08
Isovaleric acid, μmoL/g	2.21	2.05	2.23	0.19	0.77
Valeric acid, μmoL/g	0.49	0.47	0.51	0.07	0.92
Total SCFAs, μmoL/g	50.17	49.76	49.11	2.27	0.94

## Data Availability

The data presented in this study are available in the article.

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
