# Peer review of "Partial Substitution of Whey Protein Concentrate with Spray–Dried Porcine Plasma or Soy Protein Isolate in Milk Replacer Differentially Modulates Ileal Morphology, Nutrient Digestion, Immunity and Intestinal Microbiota of Neonatal Piglets"

_animals, 2023, doi:10.3390/ani13213308_

Round 1

Reviewer 1 Report

This paper studied the effects of different protein sources in suckling pig formula milk powder on intestinal morphology and function, digestive ability, immunity, and intestinal microflora of suckling piglets, and confirmed the feasibility of partially replacing WPC with SDPP as a protein source for suckling pig milk powder. However, several issues in the manuscript should be noted and modified.

1.     Is the nutritional level a calculated or a tested value? The composition and content of Premix should be presented.

2.     If the final body weight is recorded, why are the relevant results not shown? It is highly recommended to show data on growth performance, such as ADG, ADFI and F/G ratio.

3.     2.5 Digestive activities? Maybe you meant digestive enzyme activity? The grammar of the manuscript needs improvement.

4.     In the article, it is repeatedly stated that SDPP is feasible as the main protein source in the formula milk powder, but the formula of SDPP group only contains 6 % SDPP, WPC is still the main protein source in the SDPP diet, so the statement needs to be modified.

5.     Line 139, what samples?

6.     Figure 6, the text in the image is blurry, please adjust.

The grammar of the manuscript needs improvement.

Author Response

We would like to thank you for giving us an opportunity to revise our manuscript. We appreciate editors and reviewers for their positive and constructive comments and suggestions on our manuscript entitled “Effects of formula protein sources on ileal morphology, nutrients digestion, immunity and gut microbiota of neonatal piglets”. Those comments are all valuable and very helpful for improving this paper. All changes made to the text are in red color. The point-by-point response to the comments and suggestions are listed as below. We hope the new manuscript will meet the requirements. Looking forward to hearing from you!

3. Point-by-point response to Comments and Suggestions for Authors

Comments 1: Is the nutritional level a calculated or a tested value? The composition and content of Premix should be presented.

Response 1: Thank you for pointing this out. The nutritional level is a calculated value. We have added the composition and content of Premix in this version. (Table 1)

Comments 2: If the final body weight is recorded, why are the relevant results not shown? It is highly recommended to show data on growth performance, such as ADG, ADFI and F/G ratio.

Response 2: Thanks for your suggestion! Because the study was originally designed to focus on the impact of protein source of milk replacer on the intestinal health of neonatal piglets. Then, we have added the growth performance and diarrhea index in this version. (Table 3 and Figure 1)

Comments 3: 2.5 Digestive activities? Maybe you meant digestive enzyme activity? The grammar of the manuscript needs improvement.

Response 3: Thanks for your suggestion! We have replaced ‘Digestive activities’ by ‘digestive enzyme activity’ throughout manuscript. (L422)

Comments 4: In the article, it is repeatedly stated that SDPP is feasible as the main protein source in the formula milk powder, but the formula of SDPP group only contains 6 % SDPP, WPC is still the main protein source in the SDPP diet, so the statement needs to be modified.

Response 4: Yes, the WPC is still the main protein source in either SDPP or SPI group, considering the dairy-based protein WPC is commonly used as the main protein source in milk replacer for piglets as before[1], however, we intended to observe the growth and health responses of piglets to SDPP( recognized as a bioactive stuff-enriched protein source), or SPI (typical plant protein source in milk replacer but cheaper). And referring SDPP used as 2~4% in weaning diet[2,3], we used 6% of SDPP as isonitrogenous replacement. Also, we modified the description in this version. [the milk replacer with 17.70 % WPC (WPC group, n=8), the milk replacer with 6% SDPP isonitrogenously substituting WPC (SDPP group, n=8), and the basal diet with 5.13% SPI isonitrogenously substituting WPC (SPI group, n=8)]. Meanwhile, considering that SDPP and SPI are partially substituting WPC, we also modified the description of the article title.

Comments 5: Line 139, what samples?

Response 5: Sorry for the unclear description! This ‘sample’ refers to the colon chyme sample. We have modified this content. (L466)

Comments 6: Figure 6, the text in the image is blurry, please adjust.

Response 6: Sorry about this! We have modified this content. (Figure 7)

4. Response to Comments on the Quality of English Language

Point 1: The grammar of the manuscript needs improvement.

Response 1: Yes, we have tried our best to improve the English quality.

References above mentioned

  1. Navis M.; Muncan V.; Sangild P.T.; Willumsen L.M.; Koelink P.J.; Wildenberg M.E.; Abrahamse E.; Thymann T.; van Elburg R.M.; Renes I.B. Beneficial Effect of Mildly Pasteurized Whey Protein on Intestinal Integrity and Innate Defense in Preterm and Near-Term Piglets. Nutrients 2020, 12, 1125-1126.
  2. Kats, L.J.; Nelssen, J.L.; Tokach, M.D.; Goodband, R.D.; Hansen, J.A.; Laurin, J.L. The effect of spray-dried porcine plasma on growth performance in the early-weaned pig. Journal of Animal Science 1994, 2075.
  3. Müller, L.K.F.; Paiano, D.; Gugel, J.; Lorenzetti, W.R.; Santurio, J.M.; Fernando, D.C.T.; Da Gloria, E.M.; Baldissera, M.D.; Da Silva, A.S. Post-weaning piglets fed with different levels of fungal mycotoxins and spray-dried porcine plasma have improved weight gain, feed intake and reduced diarrhea incidence. Microbial Pathogenesis 2018, S0882401017310045.

Reviewer 2 Report

title and abstract: the term formula is not correct. It gives the impression to the reader that a manuscript is about the test of a human milk replacer. Use „milk replacer” and supplementary milk feeding instead.

line 13 and 19: same sentence, change it

line 42: breastfeeding term is not used in animal science. Use nurse instead.

line 44: change term formula to milk replacer.

line 53: infant formula means milk replacer for human babies. Use piglets or neonates.

lines 54-55: This is completely misleading. Piglets and babies not intolerant to lactose. The introduction and the aim of the study has to be clear: is this experiment is the test of different human milk replacer or protein sources for piglet milk replacer? The whole introduction needs to be corrected based on that.

line 70: number of animal experimental approval has to be given.

line 88: this is contradictory. previously the feeding times were given, and here it states ad libitum access. The automatic feeder, and its mode of work needs to be described in detail. Mixing ratios of milk replacer with water needs to be given and temperature as well.

Table 1: The authors stating previously, that they purchased the milk replacers from different companies. How they have the exact composition, and having nearly identical nutrient content. Hard to believe in this.

Give manufacturer and brand name of components in footnote. Provide nutrient content of premix.

line 92: drink is for humans in the pub. Animals has access to water. What feed means here? Did the piglets have access to solid feed? This has to be described in detail.

line 98: for what?

line 105: give developer city and country

line 108: what concentration of solution?

line 113: manufacturer, city, country missing, correct everywhere

line 139: sample of what?

line 147: the observations of piglets’ health status and the findings are missing. I have been raising personally 2 day old piglets with milk replacer. One thing is sure, due to the abrupt feed change they will exhibit diarrhea. This affects the results, needs to be described in detail.

line 148: city, country missing

Figure 1 and all others: use superscript letters to demonstrate significant differences. Lines over bars and starts are not correct, they suggest pairwise comparison. Use SAS type of explanation.

Table 3. provide resolution for treatment codes as well below the table.

Figure 5 and 6: legend and text invisible small.

line 219: in which species is true?

discussion: discussion is very weak compared to the variety of data presented.

The language itself is good, I have concerns with the term formula. In my opinion it is misleading, I suggest to use milk replacer.

Author Response

We would like to thank you for giving us an opportunity to revise our manuscript. We appreciate editors and reviewers for their positive and constructive comments and suggestions on our manuscript entitled “Effects of formula protein sources on ileal morphology, nutrients digestion, immunity and gut microbiota of neonatal piglets”. Those comments are all valuable and very helpful for improving this paper. All changes made to the text are in red color. The point-by-point response to the comments and suggestions are listed as below. We hope the new manuscript will meet the requirements. Looking forward to hearing from you!

Comments 1: line 13 and 19: same sentence, change it

Response 1: Sorry about this! We have corrected this error. (L20-L21)

Comments 2: line 42: breastfeeding term is not used in animal science. Use nurse instead.

Response 2: Sorry for the mistake! We have replaced ‘breastfeeding’ by ‘nurse’ throughout manuscript. (L106)

Comments 3: line 44: change term formula to milk replacer.

Response 3: Thanks for your suggestion! We have replaced ‘formula’ by ‘milk replacer’ throughout manuscript.

Comments 4: line 53: infant formula means milk replacer for human babies. Use piglets or neonates.

Response 4: Yes, we have modified the description in this version. (It has been reported that milk replacer with WPC contributes to the intestinal maturation and health of preterm piglets. (L116)

Comments 5: lines 54-55: This is completely misleading. Piglets and babies not intolerant to lactose. The introduction and the aim of the study has to be clear: is this experiment is the test of different human milk replacer or protein sources for piglet milk replacer? The whole introduction needs to be corrected based on that.

Response 5: Yes, we have deleted this sentence in this version. This experiment is the test of protein source of milk replacer on the growth performance, ileal morphology, nutrients digestion, immunity and gut microbiota of neonatal piglets. The whole introduction has been corrected based on that in this version.

Comments 6: line 70: number of animal experimental approval has to be given.

Response 6: The number of animal experimental approval has been presented in the ‘Institutional Review Board Statement’ at the end of the article. (L849)

Comments 7: line 88: this is contradictory. previously the feeding times were given, and here it states ad libitum access. The automatic feeder, and its mode of work needs to be described in detail. Mixing ratios of milk replacer with water needs to be given and temperature as well.

Response 7: Sorry for the unclear description! We have added the detailed information in this version (L216-L220).

Comments 8: Table 1: The authors stating previously, that they purchased the milk replacers from different companies. How they have the exact composition, and having nearly identical nutrient content. Hard to believe in this.

Response 8: Thanks for your suggestion! We have given the relevant content in the footnote of Table 1 (L397-L402).

Comments 9: Give manufacturer and brand name of components in footnote. Provide nutrient content of premix.

Response 9: Thanks for your suggestion! We have given the relevant content in the footnote of Table 1 (L403-L406).

Comments 10: line 92: drink is for humans in the pub. Animals has access to water. What feed means here? Did the piglets have access to solid feed? This has to be described in detail.

Response 10: Sorry for the mistake! We have replaced ‘drink’ by ‘water’ and replaced ‘food’ by ‘milk replacer’ in this version (L408).

Comments 11: line 98: for what?

Response 11: Sorry for the unclear description! Chyme samples from the colon were collected for 16S rRNA sequencing to assess the microbial community composition (L416-L417).

Comments 12: line 105: give developer city and country

Response 12: Yes, we have added the missing information in this version (L421-L423).

Comments 13: line 108: what concentration of solution?

Response 13: Yes, we have added the missing information in this version (L426).

Comments 14: line 113: manufacturer, city, country missing, correct everywhere

Response 14: Yes, we have added the missing information in this version (L430).

Comments 15: line 139: sample of what?

Response 15: Sorry for the unclear description! This ‘sample’ refers to the colon chyme sample. We have modified this content. (L466)

Comments 16: line 147: the observations of piglets’ health status and the findings are missing. I have been raising personally 2 day old piglets with milk replacer. One thing is sure, due to the abrupt feed change they will exhibit diarrhea. This affects the results, needs to be described in detail.

Response 16: Thanks for your suggestion! The data about growth performance and diarrhea index were presented in Table 3 and Figure 1. Because the study was originally designed to focus on the impact of protein source of milk replacer on the intestinal health of neonatal piglets. Then, we have added content on the growth performance and diarrhea index in this version.

Comments 17: line 148: city, country missing

Response 17: Yes, we have added the missing information in this version (L481).

Comments 18: Figure 1 and all others: use superscript letters to demonstrate significant differences. Lines over bars and starts are not correct, they suggest pairwise comparison. Use SAS type of explanation.

Response 18: Thanks for your suggestion! We have used superscript letters to demonstrate significant differences in this version.

Comments 19: Table 3. provide resolution for treatment codes as well below the table.

Response 19: Thanks for your suggestion! We have added the resolution for treatment codes in the footnote of Table 4 (L579-L581).

Comments 20: Figure 5 and 6: legend and text invisible small.

Response 20: Yes, we have modified this content. (Figure 6 and 7)

Comments 21: line 219: in which species is true?

Response 21: Sorry for the unclear description! What we want to express is that the efficiency of partial substitution of WPC by animal-based or plant-based protein on the growth and health of piglets needs to be determined. (L641-L643)

Comments 22: discussion: discussion is very weak compared to the variety of data presented.

Response 22: Thanks! we have modified the discussion section and added a discussion about the growth performance and diarrhea index in this version. (L644-L652)

4. Response to Comments on the Quality of English Language

Point 1: The language itself is good, I have concerns with the term formula. In my opinion it is misleading, I suggest to use milk replacer.

Response 1: Thanks for your suggestion! We have replaced ‘formula’ by ‘milk replacer’ throughout manuscript.

Round 2

Reviewer 1 Report

Manuscript can be accepted.

Author Response

Response 1: Thank you for your help in revising and publishing the article.

Reviewer 2 Report

The manuscript improved significantly. I only miss more intensive discussion of the results. Some minor issues detected:

Title and abstracts: It is necessary to write out treatment codes in title, and also in first mention in abstracts. These sections may read separately, and they have to be self understandable.

line 20: delete: of important

keywords: add:    milk replacer; replace swine by piglets

line 92: (ADG)and    space missing

line 96: I would change brest milk to sow milk

line 109: vitamines with number in their name, number should be subscript

line 116:what kind of anti-coagulant containing tubes were used?

line 194 and under every table and figure where relevant: ab Means marked with the same letter do not differ (P>0,05)   instead of a,b Means P < 0.05.

The English is fine, some little things caught my eyes. 

Author Response

We would like to thank you for giving us an opportunity to revise our manuscript. We appreciate editors and reviewers for their positive and constructive comments and suggestions on our manuscript entitled “Effects of formula protein sources on ileal morphology, nutrients digestion, immunity and gut microbiota of neonatal piglets”. Those comments are all valuable and very helpful for improving this paper. All changes made to the text are in red color. The point-by-point response to the comments and suggestions are listed as below. We hope the new manuscript will meet the requirements. Looking forward to hearing from you!

3. Point-by-point response to Comments and Suggestions for Authors

Comments 1: Title and abstracts: It is necessary to write out treatment codes in title, and also in first mention in abstracts. These sections may read separately, and they have to be self understandable.

Response 1: Thanks for your suggestion! We have added them.

Comments 2: line 20: delete: of important

Response 2: Yes, we have deleted “of important” in this version. (L21)

Comments 3: keywords: add: milk replacer; replace swine by piglets

Response 3: We have added “milk replacer” and replaced “swine” by “piglets” in keywords. (L43)

Comments 4: line 92: (ADG) and space missing

Response 4: Sorry for the mistake! We have added the space in this version. (L94)

Comments 5: line 96: I would change brest milk to sow milk

Response 5: Thanks for your suggestion! We have replaced “breast milk” by “sow milk” in this version. (L98)

Comments 6: line 109: vitamines with number in their name, number should be subscript

Response 6: Thanks for your suggestion! We have subscripted the numbers. (L111)

Comments 7: line 116: what kind of anti-coagulant containing tubes were used?

Response 7: Sorry for the unclear description! What we want to express is that plasma samples were obtained by centrifuging blood samples stored in heparin anticoagulated tubes at 3000 × g for 15 min at 4°C. (L118)

Comments 8: line 194 and under every table and figure where relevant: ab Means marked with the same letter do not differ (P>0,05)   instead of a,b Means P < 0.05.

Response 8: Thanks for your suggestion! We have replaced “a,b Means P < 0.05” by “ab Means marked with the same letter do not significantly differ (P>0,05)” throughout manuscript.

4. Response to Comments on the Quality of English Language

Point 1: The English is fine, some little things caught my eyes.

Response 1: Yes, we have tried our best to improve the English quality.